# *Bcl*-*xL* Promotes the Survival of Motor Neurons Derived from Neural Stem Cells

**DOI:** 10.3390/biology12010132

**Published:** 2023-01-13

**Authors:** Yunqin Wu, Xiaohua Peng, Song Ang, Yue Gao, Yue Chi, Jinling Wang, Chengcheng Tang, Xiaoqing Zhou, Yanxian Feng, Kun Zhang, Qingjian Zou, Min Chen

**Affiliations:** 1Guangdong Provincial Key Laboratory of Large Animal Models for Biomedicine, South China Institute of Large Animal Models for Biomedicine, Wuyi University, Jiangmen 529020, China; 2International Healthcare Innovation Institute (Jiangmen), Jiangmen 529040, China

**Keywords:** *Bcl*-*xL*, neural stem cells, motor neurons, survival

## Abstract

**Simple Summary:**

Neural stem cells (NSCs) are used for developing cellular replacement strategies for motor neuron diseases. However, this approach has been limited by poor survival and the insufficient function of NSC-derived motor neurons. Seeking to overcome these challenges, we generated pro-survival and drug-inducible NSCs, called *NILB*-mNSCs, which limited the neuronal death of motor neurons in a medium with doxycycline in vitro. The data obtained indicate that the special genetic modification of NSCs will be helpful for survival and the formation of motor neurons.

**Abstract:**

Neural stem cell (NSC) transplantation creates new hope for the treatment of neurodegenerative disorders by direct differentiation into neurons. However, this technique is limited by poor survival and functional neuron deficiency. In this research study, we generated pro-survival murine NSCs (mNSCs) via the ectopic expression of *Bcl*-*xL*. A doxycycline (Dox)-inducible *Ngn2-Isl1-Lhx3* system was also integrated into the mNSC genome. The four gene-modified mNSCs can rapidly and effectively differentiate into motor neurons after Dox treatments. Ectopic *Bcl*-*xL* could resist replating-induced stress, glutamate toxicity, neuronal apoptosis and remarkably promote the survival of motor neurons. Taken together, we established genetically modified mNSCs with improved survival, which may be useful for motor neuron degenerative diseases.

## 1. Introduction

Selective motor neuron loss is the most common and striking pathological characteristic observed in the brainstem, spinal cord and motor cortex of patients with motor neuron disease [1]. Stem cells can be differentiated into almost any cell type of the human body, providing great potential for the replenishment of the neuronal population lost in motor neuron diseases. Thus, intensive effort has been directed toward the development of stem cell-based strategies for motor neuron diseases. Both adult and embryonic stem cells are studied in animal models of amyotrophic lateral sclerosis (ALS), and at least two types of autologous cells (bone marrow mononuclear cells and mesenchymal stem cells) are undergoing early-stage clinical trials [2,3,4,5]. Neural stem cells (NSCs) are an attractive population for neurological repair, because they can be isolated and maintained by available protocols, greatly expanded in culture and differentiated into specific subtype neurons [6]. Several previous studies demonstrated that NSCs transplantation improved motor function in different neurodegenerative disease animal models, including ALS [7], chronic spinal cord injury [8], Parkinson’s disease [9] and so on. Nevertheless, there are still many challenges to be faced at the biological cell and neuroscience system levels. First, NSCs are more likely to differentiate into glial cells rather than functional neurons after transplantation in vivo, which is a disadvantage for neuron replacement therapy with respect to neurodegenerative diseases. Tashiro et al. showed that more than 80% NSCs differentiated to glia cells after transplantation in mice with chronic spinal cord injury [8]. Moreover, mass death and the functional deficiency of transplanted cells represent considerable therapeutic hurdles. Ziavra et al. demonstrated that the percentage of surviving NSCs in the host striatum was extremely low (0.2–0.6%), resulting in limited cell replenishment [9]. Graft cell death was initiated by three principal pathways: immunorejection, which involves donor tissue recognition and destruction by allo-specific immune cells of the recipient [10]; anoikis due to the anchorage-dependent cells detachment from their substrate for injection [11]; and inflammation-related microenvironment, such as free radicals, cytokines and natural killer cells in the pathological site of neurodegenerative diseases [12].

In response to these challenges, highly pure motor neurons were generated using an inducible system for directed differentiation, and this system regulates *Neurogenin2* (*Ngn2*), *insulin gene enhancer 1* (*Isl1*) and *LIM homeobox 3* (*Lhx3*) expression [13]. The anti-apoptosis gene, *Bcl*-*xL*, that limits the death of motor neurons was then transferred into these cells. These directionally inducible and pro-survival mNSCs were converted into motor neurons rapidly and efficiently. The resulting induced neurons also resisted neuronal death induced by replating procedures and glutamate toxicity. These *Bcl*-*xL* overexpression motor neurons can fire repetitive action potentials and express voltage-gated ion channels.

## 2. Materials and Methods

### 2.1. Murine Neural Stem Cell Culture

All cell culture reagents were purchased from Gibco except for the specifically mentioned reagents.

Cells in the cortex of E13.5 C57BL/6 mouse pups were dissociated by incubation for 7 min at 37 °C in 0.25% trypsin. The dissociated cells were centrifuged for 10 min at 200× *g* and then washed with 1× DPBS. The cell suspension was centrifuged for 10 min at 200× *g* and plated on a Corning Matrigel-coated 6-well plate in mNSC medium, which comprised neurobasal medium, 1× B27, 1× Glutamax and 0.5× penicillin/streptomycin, and then supplemented with 20 ng/mL FGF and 20 ng/mL EGF (Pepro Tech, Cranbury, NJ, USA). The collected mNSCs were characterized by staining using antibodies against NSC markers Pax6 and Sox2 [14].

### 2.2. Plasmid Construction

The PB-*Ngn2*-*Isl1*-*Lhx3* (PB-*NIL*) plasmid was a gift from Prof. Alessandro Rosa (Center for Life Nano Science, Rome, Italy) [13]. The PB-*Bcl*-*xL*-*Luciferase*-*GFP* (PB-*BLG*) plasmid was generated as previously described [15].

### 2.3. Generation of Stable mNSCs

mNSCs were co-transfected with transposable vector and the PiggyBac transposase (PB-*NIL*: transposase = 4:1) using Lipofectamine™ 3000 (Invitrogen, Carlsbad CA, USA). Three days post-transfection, cells that were selected with 5 μg/mL blasticidin gave rise to stable cell lines (termed *NIL*-mNSCs). After selection, *NIL*-mNSCs cells were re-transfected with PB-*BLG* (PB-*BLG*: transposase = 4:1) and then sorted by fluorescence-activated cell sorting (FACS) based on the expression of GFP, generating directionally inducible and pro-survival mNSCs (termed *NILB*-mNSCs).x

### 2.4. Luciferase Assay

*NIL*-mNSCs and *NILB*-mNSCs were seeded in 96-well plates at a density of 5000 cells/well for 48 h. Luciferase activities were determined according to the protocol provided by the manufacturer (Yeasen). Briefly, after 48 h, the medium was removed from the cells and replaced with fresh media containing D-luciferin (0.3 mg/mL) and then incubated at 37 °C for 10 min. After incubation, the absorbance at 560 nm was measured on a multimode reader (LB943, Berthold Technologies, Bad Wildbad, Germany).

### 2.5. Motor Neuron Differentiation

*NIL*-mNSCs and *NILB*-mNSCs were seeded onto Matrigel-coated culture vessels (5.0 × 10^4^/well of 24-well plate) with or without glass coverslips. Differentiation was induced by adding 1 μg/mL Dox in the mNSC medium. The medium was changed every day. After 48 h of Dox induction, neuronal cultures were switched to a murine motor neuron (mMN) medium supplemented with 10 ng/mL BDNF (Alomone labs, Jerusalem, Israel), 10 ng/mL GDNF (Alomone labs) and 10 ng/mL NT3 (Alomone labs). The mMN medium was composed of F12, 5% horse serum, 1× N2, 1× B27 and 1× Glutamax.

### 2.6. Immunofluorescence

Cell cultures at the specified time points were fixed with 4% paraformaldehyde for 15 min and then permeabilized/blocked in blocking solutions (1× PBS + 0.1% Triton X-100 + 3% BSA) for 1 h at room temperature. Immunostaining was performed with rabbit anti-Pax6 (Biolegend, 1:200, San Diego, CA, USA); mouse anti-Sox2 (R&D, 1:200); mouse anti-Isl1/2 (DSHB, 1:200, Iowa City, IA, USA); rabbit anti-TUJ1 (Abcam, 1:2000, Shanghai, China); mouse anti-HB9 (DSHB, 1:100); mouse anti-MAP (Sigma, 1:200, Kanagawa, Japan) primary antibodies and anti-mouse Alexa-488/555 (1:2000, CST); and anti-rabbit Alexa-488/555 (1:2000, CST) secondary antibodies. The images were obtained using Olympus BX51 Microscope.

### 2.7. Isolation and Culture of Primary Neonatal Glial Cells

Murine glial cells cultures were prepared from 5-day-old wild-type mice as described previously [16]. Briefly, cortices were digested with 0.25% trypsin at 37 °C for 15 min. The dissociated cells were then washed with Dulbecco’s modified Eagle’s medium (DMEM) containing 10% fetal bovine serum, 1× Glutamax, 1× NEAA, 1× sodium pyruvate and 1× penicillin/streptomycin. The cell suspension was centrifuged for 10 min at 500× *g* and then plated on 10 cm dishes supplemented with DMEM, 10% fetal bovine serum, 1× Glutamax, 1× NEAA, 1× sodium pyruvate and 1× penicillin/streptomycin. The glia cells were then passaged 3 times before co-culturing with *NIL*-mNSCs or *NILB*-mNSCs.

### 2.8. Replating-Induced Stress

Equal amounts of *NIL*-mNSCs and *NILB*-mNSCs at 5.0 × 10^4^/well were planted on monolayer glia cultures prepared from newborn pups as described in Section 2.7. After 5 days of induction with Dox (1 μg/mL), the cells were dissociated with Accutase (Stem Cell Technologies) and re-plated on Matrigel-coated 24-well plates. The survival rate before replating was set as 100%, to which the subsequent surviving cells at 3 d and 7 d after replating were normalized. The surviving cells were then quantified at 200× magnification. Three non-overlapping visual fields were randomly selected per well from a 24-well plate. Three independent experiments were performed. The number of surviving neuronal cells was counted, and the average number was calculated for survival rates. Moreover, the axon length of the surviving neurons was measured using ImageJ. At least 30 neurons in 3 randomly selected fields were used for measurements, and the mean axon length was calculated.

### 2.9. Glutamate Toxicity Assay

Equal amounts of *NIL*-mNSCs and *NILB*-mNSCs at 5.0 × 10^4^/well were planted on a Matrigel-coated 24-well plate. After differentiation with Dox treatments, as described in Section 2.5, these induced motor neurons were cultured in a neurotrophin-deprived medium for 7 days. Afterwards, glutamate (20 μM, MCE) and L-trans-Pyrrolidine-2,4-dicarboxylic acid (100 μM, PDC, Sigma) were added to the medium for 7 days to induce glutamate overload toxicity [17]. Cultures were subsequently maintained for additional 7 days for the immunocytochemistry and cell counting.

### 2.10. Apoptosis Assay

Equal amounts of *NIL*-mNSCs and *NILB*-mNSCs at 5.0 × 10^4^/well were planted on a Matrigel-coated 24-well plate. After differentiation with Dox (1 μg/mL) for 5 days, these cells were switched to the mMN medium without Dox, and half of the cells were changed every other day until 14 days. Cultures were harvested and stained with annexin-V and 7-AAD according to the manufacturer’s instruction (Annexin V:PE Apoptosis Detection Kit I, BD Pharmingen). The percentages of annexin-V and 7-AAD double positive cells were quantified as apoptotic cells. The stained sample was detected by FACS.

### 2.11. Patch-Clamp Recordings

Recordings were performed using *NIL*-mNSC- and *NILB*-mNSC-derived motor neurons at 3 weeks. The pH of the bath solution containing (in mM) 140 NaCl, 5 KCl, 2 MgCl_2_, 10 HEPES and 10 glucose was adjusted to 7.4 with NaOH. Afterwards, the pH of the whole-cell pipette solution containing (in mM) 123 K-gluconate, 10 KCl, 1 MgCl_2_, 10 HEPES, 1 EGTA, 0.1 CaCl_2_, 1 K_2_ATP, 0.2 Na_4_GTP and 4 glucose was adjusted to 7.2 with KOH. Sodium currents, potassium currents and action potentials were recorded with the whole-cell configuration.

### 2.12. In Vivo Transplantation Studies

SOD1-G93A mice were purchased from The Jackson Laboratory. All animal care and experimental procedures were approved by the Ethical Committee on Animal Experiments at School of Biotechnology and Health Sciences, Wuyi University. The pups (P3–4) were cryoanesthetized on ice and received 2 injections of EGFP-positive *NIL*-mNSCs or *NILB*-mNSCs (1 × 10^5^ cells/μL/injection) using a Hamilton syringe (33G). Two days post-transplantation, the pups were administered i.p. with Dox (25 mg/kg, 4 times) every other day. After 10 days transplantation, the mice were perfusion-fixed for immunocytochemistry analyses.

### 2.13. Statistical Analysis

All statistics, including statistical tests, sample sizes and types of replicates, are described in the Figure legends. A *p* value of < 0.05 was considered to be statistically significant.

## 3. Results

### 3.1. Generation of Directionally Inducible and Pro-Survival mNSCs

Multigene-modified mNSCs were established by isolating mNSCs from the cortex of E13.5 mouse pups and culturing them as initiating cells. Primary mNSCs were cultured in FGF and EGF media, and more than 95% of the cells expressed the mNSCs markers Sox2 and Pax6 (Figure 1A). As illustrated in Figure 1B, multigene-modified mNSCs were generated by two steps. First, mNSCs were transfected with PB-*NIL* and then selected with blasticidin to obtain Dox-inducible mNSCs (termed *NIL*-mNSCs). After induction with Dox, *NIL*-mNSCs expressed transgenic protein Isl1/2 (Figure 1C). Second, *NIL*-mNSCs were re-transfected with PB-*BLG*, resulting in 65.7% GFP-positive cells (Figure 1D), which were further sorted by FACS to obtain the inducible and pro-survival mNSCs (termed *NILB*-mNSCs). In comparison with *NIL*-mNSCs, *NILB*-mNSCs showed significantly higher activities with respect to luciferase (Figure 1E).

### 3.2. Ectopic Bcl-xL Does Not Affect Motor Neuron Differentiation

The protocol of mNSC differentiation toward motor neurons is outlined in Figure 2A. *Ngn2*-*Isl1*-*Lhx3* conditional expression was induced with Dox for 2 days in the mNSC medium and 3 days in mMN medium. During this process, cells rapidly changed their morphology to condensed nuclei, long axons and multiple neuritis (Figure 2B). Remarkably, over 80% of *NIL*-mNSCs and *NILB*-mNSCs were converted to neuron-like cells and positive for the neural maker TUJ1 stained with TUJ1-positive neurons (Figure 2C,D). Moreover, both *NIL*-mNSCs and *NILB*-mNSCs differentiated into typical motor neurons, expressing spinal motor neuron marker HB9 and mature neuronal marker MAP2 (Figure 2E–H). Notably, no difference was observed in motor neuron differentiation efficiency between *NIL*-mNSCs and *NILB*-mNSCs (HB9: 59.6% vs. 55.8%; MAP2: 45.6% vs. 46.7%), indicating that *Bcl*-*xL* overexpression had no effect on neural differentiation.

### 3.3. Ectopic Bcl-xL Promotes the Survival of Induced Motor Neurons

Generally, *Bcl*-*xL* is a common anti-apoptotic protein that promotes cell survival. The pro-survival capacity of *Bcl*-*xL* was determined using two different in vitro models, namely, replating-induced stress [18] and glutamate excitotoxicity [17]. In the first model, motor neurons are sensitive to systemic stress induced by a replating procedure in vitro. Over 60% motor neurons died after replating even after being co-cultured with primary murine glial cells [18]. As shown in Figure 3A, before replating, the number of induced motor neurons of *NILB*-mNSCs present on murine glia was higher than that of *NIL*-mNSCs, although the same number of starting cells was plated, suggesting the protective effect of *Bcl*-*xL* overexpression during passage procedure. At 3 days post-replating, the survival rates of *NIL*-mNSCs and *NILB*-mNSCs were reduced to 54.94% and 49.77%, respectively, but no significant difference was observed. However, after 7 and 12 days of replating, the survival rate of *NIL*-mNSCs was statistically reduced to 25.93% and 7.41%, respectively, compared with that of *NILB*-mNSCs (37.33% and 30.10%, Figure 3A,B). In addition, a marked difference was observed in the axon length of *NILB*-mNSC-derived motor neurons compared with the *NIL*-mNSCs group (Figure 3C).

The selective loss of glutamate transporter in the brain and spinal cord of ALS patients suggested that glutamate accumulation may contribute to ALS [19]. Thus, we also examined the survival of induced motor neurons in glutamate overload models. *NIL*-mNSC-derived motor neurons showed susceptibility to glutamate (Figure 3D,E), and approximately 51.46% of induced motor neurons died after glutamate treatment. The *Bcl*-*xL* overexpression alleviated glutamate toxicity, increasing the survival rate of *NILB*-mNSCs to 70.61%. Overall, *Bcl*-*xL* overexpression conferred the motor neurons’ pro-survival ability, suggesting that the induced motor neurons are likely to survive and perform their function.

### 3.4. Motor Neurons Derived from NILB-mNSCs Exhibit Functional Membrane Properties

The induced motor neurons were cultured beyond 2 weeks. About 48.2% cells showed 7-AAD and annexin-V dual-positive in the *NIL*-mNSC group. In contrast, fewer than 12% cells were positive for both dyes in the *NILB*-mNSC group (Figure 4A,B), indicating that obvious apoptosis was detected in the *NIL*-mNSC group but not in the *NILB*-mNSC group. After 3 weeks, whole-cell patch-clamp recordings were performed to further characterize the functional membrane properties of the surviving motor neurons. As shown in Figure 4C, weak inactivating inward currents (Na^+^) and transient outward currents (K^+^) were detected in *NIL*-mNSC-induced motor neurons. No action potentials were observed in any of the *NIL*-mNSC-induced motor neurons. *NILB*-mNSC-induced motor neurons exhibited the repetitive firing of action potentials and large voltage-gated sodium/potassium currents (Figure 4D). These data indicate that the ectopic expression of *Bcl*-*xL* increased the electrophysiological maturation of *NILB*-mNSC-induced motor neurons.

### 3.5. In Vivo Survival and Direct Differentiation of NILB-mNSCs Induced by Dox

To trace cells in vivo, *NIL*-mNSCs and *NILB*-mNSCs were labelled with EGFP, respectively. mNSCs were intrathecally injected into ALS mice (SOD1-G93A), and Dox was treated every other day (Figure 5A). The grafts survived at least 10 days, and the number of GFP-positive grafts in *NILB*-mNSCs group was much more than that of the *NIL*-mNSC group (Figure 5B). Further immunostaining experiments showed that most grafts expressed neural markers TUJ1, HB9 and MAP2 (Figure 5C).

## 4. Discussion

In this study, we found that *Bcl*-*xL* promoted the long-term survival of induced motor neurons. The death of transplanted cells presents a challenge in cell therapy for neurodegenerative, cardiovascular and other diseases [20,21,22]. *Bcl*-*xL* is a well-characterized death-suppressing molecule of the *Bcl*-*2* family, which plays a critical role in preventing neuronal apoptosis induced by diverse pathologic stimuli, including cerebral ischemic and transplant microenvironments [23,24,25]. We also found that *Bcl*-*xL* promoted neuronal elongation and electrophysiological maturation. Two previous studies have also reported that *Bcl*-*xL* enhanced neuronal differentiation and neurite outgrowths from human NSCs and mouse embryonic stem cells [26,27]. *Bcl*-*xL* inhibits the caspase-dependent signal pathway, which may be responsible for neurite degeneration [28].

The generation of sufficient and specific-subtype neurons is also a common hurdle in stem-cell-based strategies for neurodegenerative diseases. Differentiation protocols for obtaining desired neuronal subtypes are often expensive, cumbersome and require a long period of time [29,30]. *Ngn2*, *Isl1* and *Lhx3* are critical for induced motor neuron differentiation [31,32]. Many researchers generated homogeneous populations of mature functional motor neurons by the forced expression of *Ngn2*, *Isl1* and *Lhx3* in human fibroblasts, embryonic stem cells (ESCs) and induced pluripotent stem cells (iPSCs) [33,34,35]. Importantly, in comparison with ESCs and iPSCs, NSCs are easily modified by various genes, helping produce more homogeneous motor neurons in shorter periods of time. Based on a previous study by De Santis et al. [13], inducible *Ngn2*, *Isl1* and *Lhx3* genes and *Bcl*-*xL* were stably integrated into mNSC’s genome by piggyBac vectors, overcoming poor transfection efficiencies. The current results demonstrated that *NILB*-mNSCs can be rapidly and efficiently converted to motor neurons. The entire procedure generated induced motor neurons in 5 days, and the resulting induced cells expressed specific marker HB9 and mature marker MAP2. This duration was faster than that in previous observations using wild-type NPCs, and those NSCs differentiated into specific neurons by small molecules [36]. Preclinical studies documented that wild-type NPCs need 2-3 weeks to form MAP^+^ neurons [26,27]. Further, *NILB*-mNSC-induced cells became more electrophysiologically functional after 3 weeks in vitro.

*NILB*-mNSC differentiated into MAP^+^ neurons with high efficiency after Dox treatments in vivo. Preclinical studies documented that after transplantation in vivo, wild-type NPCs need 1–2 months to form neurons in ALS animal models [7,37]. The motor neurons induced by our strategy are expected to replace the damaged endogenous motor neurons rapidly, which may be helpful for acute brain and spinal cord repair. Notably, the loss of grafts is caused by multiple factors, which include immunorejection, anoikis and inflammation-related factors [10,11,12]. Pro-survival strategies have been reported, including genetic modifications, cell preconditioning and the use of biomaterials [38]. Among those strategies, multigenetic modifications endow stem cells with the capacities of anti-immune rejection, anti-inflammation and increasing angiogenesis [39]. Biomaterials as a supportive matrix for the delivery of cells into the injury cavity protect transplanted stem cells from anoikis [40]. Therefore, those pro-survival cocktails could be helpful for the survival and function of grafts.

## 5. Conclusions

In this study, we generated pro-survival and drug-inducible NSCs, called *NILB*-mNSCs, which limited the neuronal death of motor neurons after directional differentiation. This suggests that genetic modification of NSCs prior to their transplantation will enhance their survival and increase their function in cell therapy bodily functions. Further studies are required to develop a pro-survival cocktail targeting key points of potential death pathways and to determine whether genetically modified NSCs are useful in vivo.

## Figures and Tables

**Figure 1 biology-12-00132-f001:**
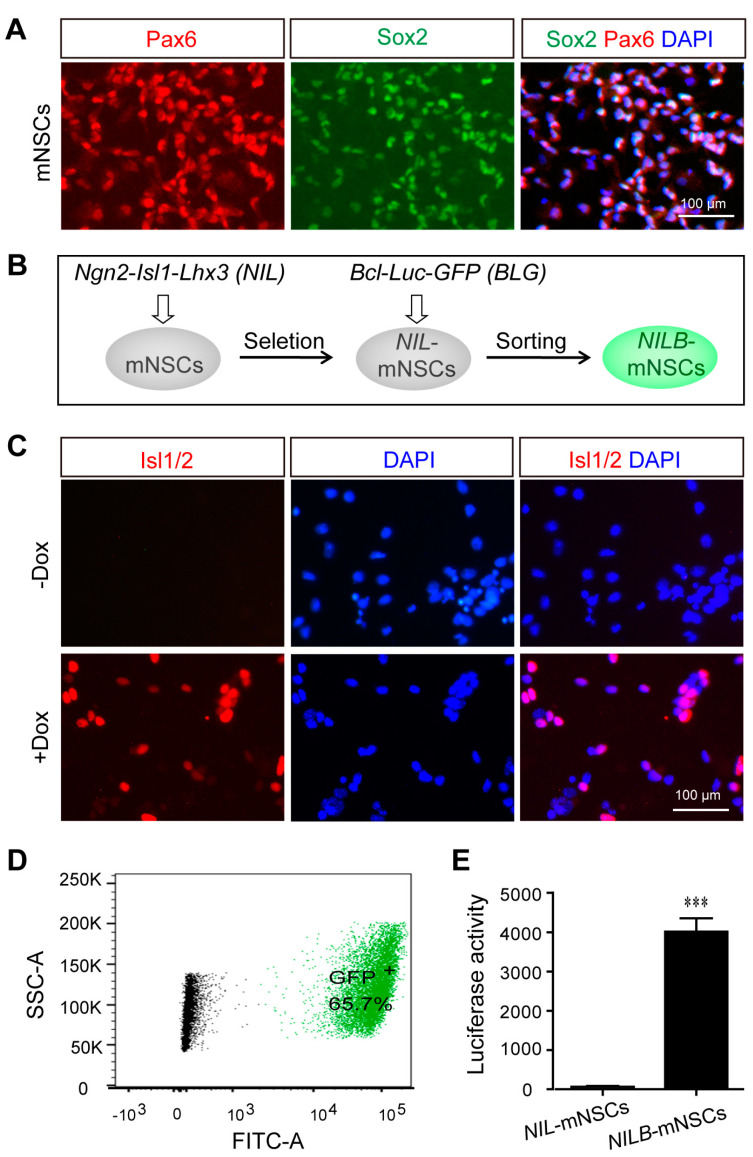
Establishment of directionally inducible and pro-survival mNSCs. (**A**) Immunostaining for the neural stem cell markers Sox2 and Pax6 in primary mNSCs. (**B**) Flow diagram of multigen-modified mNSCs lines generation. (**C**) Immunostaining for Isl1/2 protein in *NIL*-mNSCs cells treated with or without Dox. (**D**) FACS analysis of GFP expression in *NILB*-mNSCs. (**E**) Luciferase reporter assay. Error bars represent SEM; *n* = 3 separate differentiation experiments each using genetically modified *NIL*-mNSCs and *NILB*-mNSCs; two-tailed Student’s *t*-test; *** *p* < 0.001.

**Figure 2 biology-12-00132-f002:**
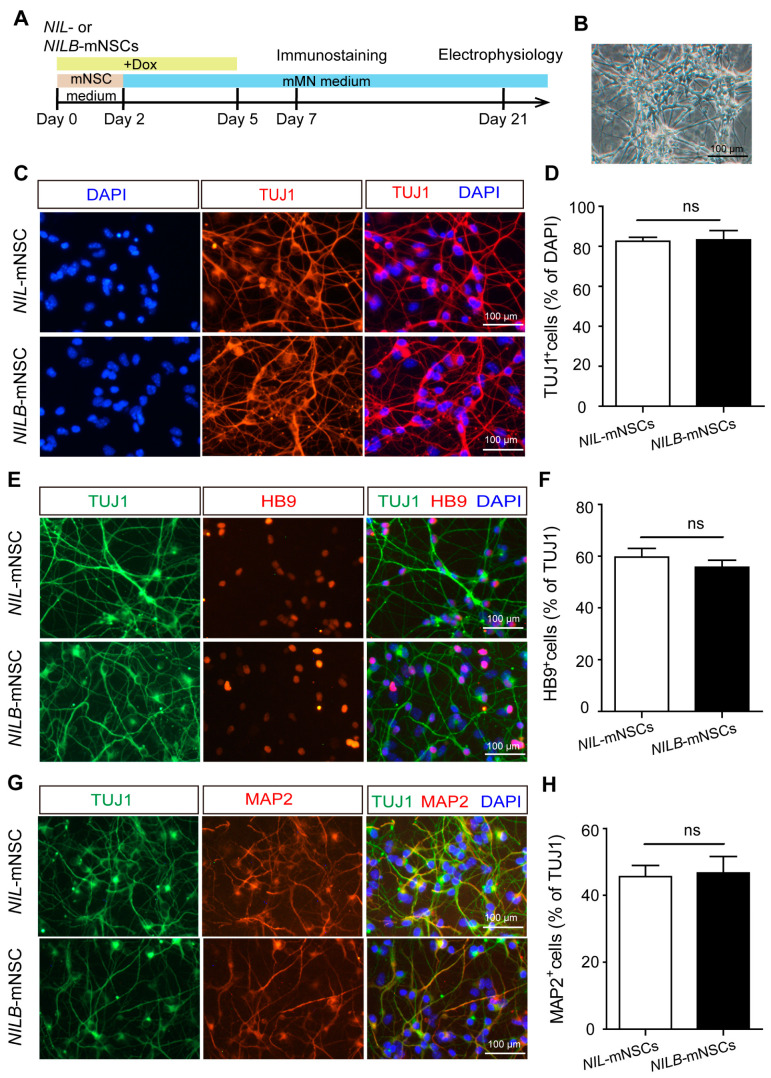
Motor neuron differentiation. (**A**) Schematic representation of the differentiation protocol. (**B**) Representative image of neuronal differentiation on day 5 of motor neuron induction, bar = 100 μm. (**C**) Immunostaining for neuronal marker TUJ1, scale bar = 100 μm. (**D**) Cell count quantification of TUJ1^+^ neurons among the total DAPI-labeled cells. Error bars represent SEM, *n* = 3 separate differentiation experiments each using the genetically modified *NIL*-mNSCs and *NILB*-mNSCs, two-tailed Student’s *t*-test; ns, not significant. (**E**) Immunostaining for motor neuron marker HB9, scale bar = 100 μm. (**F**) Cell count quantification of HB9^+^ motor neurons among the total TUJ1-labeled neurons. Error bars represent SEM; *n* = 3 separate differentiation experiments each using the genetically modified *NIL*-mNSCs and *NILB*-mNSCs; two-tailed Student’s *t*-test, ns, not significant. (**G**) Immunostaining for mature neuronal marker MAP2, scale bar = 100 μm. (**H**) Cell count quantification of MAP2^+^ motor neurons among the total TUJ1-labeled neurons. Error bars represent SEM; *n* = 3 separate differentiation experiments each using the genetically modified *NIL*-mNSCs and *NILB*-mNSCs; two-tailed Student’s *t*-test; ns, not significant.

**Figure 3 biology-12-00132-f003:**
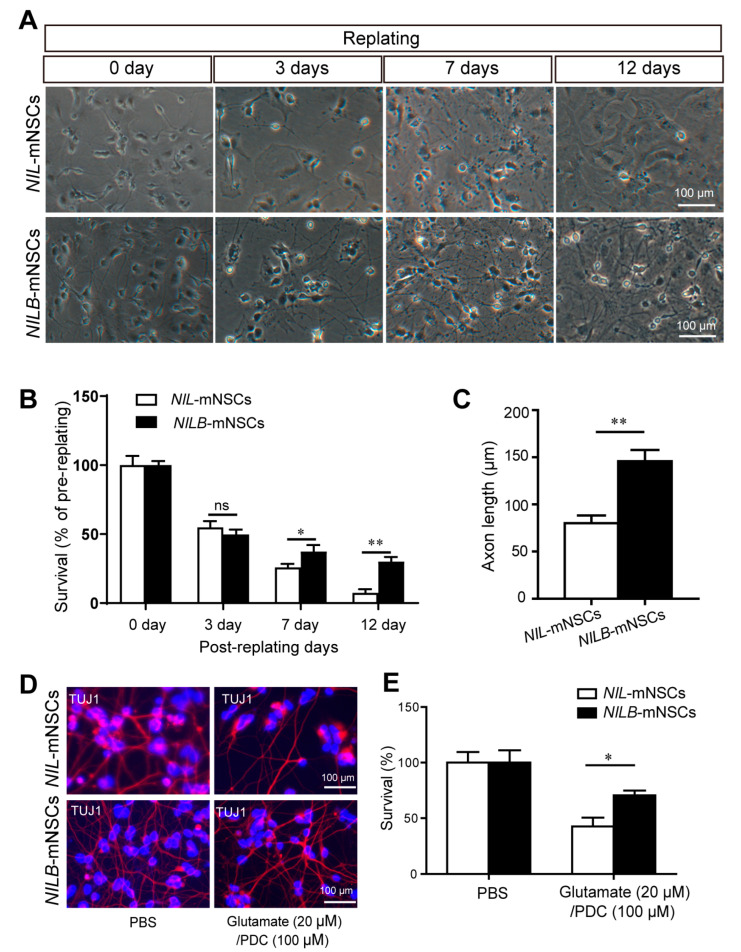
*Bcl*-*xL* overexpression promotes the survival of induced motor neurons. (**A**) Bright field for the surviving neurons pre-/post-replating, scale bar = 100 μm. (**B**) Quantitative analysis of the survival rates. Error bars represent SEM; *n* = 3 separate differentiation experiments each using the genetically modified *NIL*-mNSCs and *NILB*-mNSCs; two-tailed Student’s *t*-test; ns, not significant; * *p* < 0.05; ** *p* < 0.01. (**C**) Quantitative analysis of the axon length. Error bars represent SEM; *n* = 3 separate differentiation experiments each using the genetically modified *NIL*-mNSCs and *NILB*-mNSCs; two-tailed Student’s *t*-test; ** *p* < 0.01. (**D**) Immunostaining for the survival neurons with or without glutamate treatment, scale bar = 100 μm. (**E**) Quantitative analysis of the survival rates of induced motor neurons. Error bars represent SEM; *n* = 3 separate differentiation experiments each using the genetically modified *NIL*-mNSCs and *NILB*-mNSCs; two-tailed Student’s *t*-test; * *p* < 0.05.

**Figure 4 biology-12-00132-f004:**
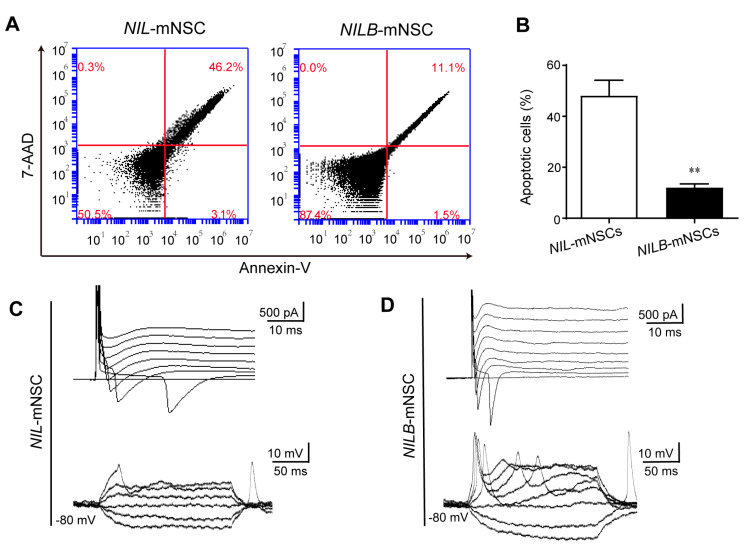
Motor neurons derived from *NILB*-mNSCs exhibit functional membrane properties in prolonged culture. (**A**) Apoptosis analysis of *NIL*-mNSCs and *NILB*-mNSCs by FACS after differentiation for 2 weeks. (**B**) Quantitative analysis of apoptotic cells. Error bars represent SEM; *n* = 3 separate differentiation experiments each using the genetically modified *NIL*-mNSCs and *NILB*-mNSCs; two-tailed Student’s *t*-test, ** *p* < 0.01. (**C**) Representative traces of voltage-dependent ion currents and action potentials recorded in *NIL*-mNSC-derived motor neurons. (**D**) Representative traces of voltage-dependent ion currents and action potentials recorded in *NILB*-mNSC-derived motor neurons.

**Figure 5 biology-12-00132-f005:**
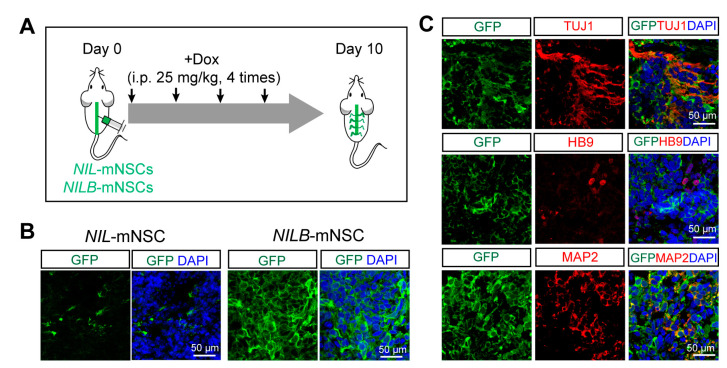
Differentiation of *NILB*-mNSCs after transplantation in SOD-G93A mice. (**A**) Flow diagram of *NIL*-mNSCs and *NILB*-mNSCs transplantation and detection. (**B**) Immunostaining for GFP-positive cells after *NIL*-mNSCs and *NILB*-mNSCs transplantation for 10 days. (**C**) Immunostaining for neuronal markers TUJ1, HB9 and MAP2 in the spinal cords of *NILB*-mNSCs-transplanted animals, bar = 50 μm.

## Data Availability

The raw data supporting the findings of this manuscript are available from Min Chen (cmin0501@wyu.edu.cn) upon reasonable request from any qualified researcher.

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
