# Peer review of "Bcl-xL Promotes the Survival of Motor Neurons Derived from Neural Stem Cells"

_biology, 2023, doi:10.3390/biology12010132_

Round 1

Reviewer 1 Report

In this manuscript Wu et al. proposed a method (Chen et al, 2022) using  inducible expression of Ngn2-Isl1-Lhx3 to accelerate motor neuron induction and Bcl-xL overexpression to block neuron apoptosis. The Ngn2-Isl1-Lhx3 were well known by directly inducing motor neuron in vitro and in vivo, and Bcl-xL, an apoptosis resistant factor, were also found play important role during neuronal development. While this paper is the first time to combine these two strategies and demonstrated Bcl-xL promotes both survival and maturation of mouse motor neuron. While the in vivo experiments manifest that NILB-mNSCs cannot be engrafted into mouse spinal cord 20 days after injection.

Generally, the paper provides evidence of using Ngn2-Isl1-Lhx3 plus Bcl-xL could stable acquire functional neuron. While there still have some problems, which must be solved before it is considered for publication. If the following problems are well-addressed, this reviewer believes that the essential contribution of this paper are valuable for the field.

Major points

1. The mNSC separated from E13.5 mouse brain are purity or contained with some mature neuron? The generation of  mNSCs is fundamental, and it would be better if you could provide more data to show the purity of mNSCs . 

2. Bcl-xL is a caspase inhibitor, it is known too low case caspase activity would inhibit neuron elongation, do you think is the high expression of Bcl-xL will stop the neuron enlogation? Can you show NIL-motor neuron’s caspase activity and prove it would not inhibit neuron elongation in your system? The authors need to prove or discuss this in the manuscript.

3. Motor neurons are diverse in terms of their morphology, connectivity, and functional properties, how similar between the NIL induced motor neuron and the real motor neuron? A deeper comparison of NILB-mNSCs derived motor neurons with published datasets of mouse motor neurons can help understanding the function of Bcl-xL. Just patch clamp result is not enough to make the conclusion that the NILB-mNSCs is more mature than NIL-mNSCs. 

4. Another or more mNSCs from different mouse strain might improve your conclusion. 

5. Is there any survival NILB-mNSCs at early period after injection (something like 2 days, 5 days or 10 days)? More attempts and data are needed to systemically investigate the in vivo phonotypes, otherwise remove the whole part of in vivo experiments which while ruins the novelty of the whole article.

6. More discussion is need especially in terms of the in vivo injection experiments.

Minor points:

1. In page 7 part 3.3, in the determination of apoptosis rate, only the results of the day 7 showed differences, which I think is not rigorous enough. Probably the difference is a transient change in the short term due to the differences in the culture conditions and so on. It would be more convincing to continue the culture for 5-7 days after the Day 7 and obtain a series of down-regulated data with differences.

2. In page 8 figure 3, the author may mismatch bar and the symbols. Is the white bar should stand for NIL-mNSC, while the black bar should stand for NILB-mNSC?

Author Response

Dear reviewer,

We appreciate very much for the careful reading of our manuscript and valuable suggestions. We have carefully considered the comments and have revised the manuscript accordingly.

The following is a point-by-point response to comments, questions and suggestions raised by reviewers.

Comments and Suggestions for Authors

In this manuscript Wu et al. proposed a method (Chen et al, 2022) using  inducible expression of Ngn2-Isl1-Lhx3 to accelerate motor neuron induction and Bcl-xL overexpression to block neuron apoptosis. The Ngn2-Isl1-Lhx3 were well known by directly inducing motor neuron in vitro and in vivo, and Bcl-xL, an apoptosis resistant factor, were also found play important role during neuronal development. While this paper is the first time to combine these two strategies and demonstrated Bcl-xL promotes both survival and maturation of mouse motor neuron. While the in vivo experiments manifest that NILB-mNSCs cannot be engrafted into mouse spinal cord 20 days after injection.

Generally, the paper provides evidence of using Ngn2-Isl1-Lhx3 plus Bcl-xL could stable acquire functional neuron. While there still have some problems, which must be solved before it is considered for publication. If the following problems are well-addressed, this reviewer believes that the essential contribution of this paper are valuable for the field.

Major points

  1. The mNSC separated from E13.5 mouse brain are purity or contained with some mature neuron? The generation of mNSCs is fundamental, and it would be better if you could provide more data to show the purity of mNSCs .

Response: As we know, mature neurons have no ability of proliferation. The mNSCs separated from E13.5 mouse brain were cultured for more than 4 passages after transgene. All of the cells showed Pax6 and Sox2 positive which were NSC specific markers (Figure 1A), indicating high purity of mNSCs in our research.

  1. Bcl-xL is a caspase inhibitor, it is known too low case caspase activity would inhibit neuron elongation, do you think is the high expression of Bcl-xL will stop the neuron enlogation? Can you show NIL-motor neuron’s caspase activity and prove it would not inhibit neuron elongation in your system? The authors need to prove or discuss this in the manuscript.

Response: In this study, we found Bcl-xL promoted neuronal elongation (Figure 3C). Two previous studies have also reported that Bcl-xL enhanced neuronal differentiation and neurite outgrowths from human NSCs and mouse embryonic stem cells [PMID: 14749429, PMID: 17673921]. Conversely, Bax(pro-apoptotic member of Bcl-2 family proteins)-dependent caspase-6 activation would result in neurite degeneration [PMID: 33801158]. We have discussed this phenomenon in the revised manuscript as follow.

“We also found Bcl-xL promoted neuronal elongation and electrophysiological maturation. Two previous studies have also reported that Bcl-xL enhanced neuronal differentiation and neurite outgrowths from human NSCs and mouse embryonic stem cells [PMID: 14749429, PMID: 17673921]. Bcl-xL inhibits the caspase dependent signal pathway which may responsible for neurite outgrowths [PMID: 33801158].”

  1. Motor neurons are diverse in terms of their morphology, connectivity, and functional properties, how similar between the NIL induced motor neuron and the real motor neuron? A deeper comparison of NILB-mNSCs derived motor neurons with published datasets of mouse motor neurons can help understanding the function of Bcl-xL. Just patch clamp result is not enough to make the conclusion that the NILB-mNSCs is more mature than NIL-mNSCs.

Response: Thanks for the reminding. It is right that just patch clamp result is not enough to make the conclusion that the NILB-mNSCs is more mature than NIL-mNSCs. To be more precise, the description of “Bcl-xL promote maturation” were removed in the revised manuscript.

  1. Another or more mNSCs from different mouse strain might improve your conclusion.

Response: Thanks for the suggestions. Our mNSCs were isolated from C57BL/6 mouse strain. In further, the disease models such as ALS mice or SMN mice will be used for the source of mNSCs, which are more suitable for cell therapy.

  1. Is there any survival NILB-mNSCs at early period after injection (something like 2 days, 5 days or 10 days)? More attempts and data are needed to systemically investigate the in vivo phonotypes, otherwise remove the whole part of in vivo experiments which while ruins the novelty of the whole article.

Response: Thanks for the reviewer’s suggestions. We have removed the whole part of in vivo experiments as suggested.

  1. More discussion is need especially in terms of the in vivo injection experiments.

Response: The issue of cell survival and functions in vivo is now been discussed in the revised manuscript as follow.

“Despite these beneficial potentials of NILB-mNSCs in vitro, the survival and functions of NILB-mNSCs after transplantation in vivo are the most important challenge. Notably, initial loss of graft cells was multifactorial in origin, including immunorejection, anoikis and inflammation-related factors [PMID:PMC2529073, 12518889 and 17721512]. Single interventions had little impact on extending survival of graft cells. The major limitation of our current study is that over express single anti-apoptosis gene Bcl-xL in mNSCs, which has largely failed to extend the survival of these mNSCs after transplantation in vivo. Accordingly, further studies are required to develop a pro-survival cocktail targeting key points of potential death pathways, and determine whether this mNSCs is useful in vivo. Pro-survival strategies have been reported, including genetic modifications, cell preconditioning, and use of biomaterials [PMID: 32518349]. Among them, multigenetic modifications can endow the stem cells with universal capacities, including anti-immune rejection, anti-inflammation, and increasing angiogenesis [PMID: 18031863]. Biomaterials as a supportive matrix for the delivery of cells into the injury cavity protect transplanted stem cells from anoikis [PMCID: PMC4157751]. Therefore, those pro-survival cocktail could be helpful for survival and function of graft.”

Minor points:

  1. In page 7 part 3.3, in the determination of apoptosis rate, only the results of the day 7 showed differences, which I think is not rigorous enough. Probably the difference is a transient change in the short term due to the differences in the culture conditions and so on. It would be more convincing to continue the culture for 5-7 days after the Day 7 and obtain a series of down-regulated data with differences.

Response: The representative neuron images on day12 post-replating and quantification of survival rate have now been included in Figures 3A-B.

  1. In page 8 figure 3, the author may mismatch bar and the symbols. Is the white bar should stand for NIL-mNSC, while the black bar should stand for NILB-mNSC?

Response: We are sorry for the mistake. Here, labels of NIL-mNSCs and NILB-mNSCs have been corrected in Figure 3B, 3C and 3E in the revised manuscript.

Reviewer 2 Report

The authors established ectopic expression of Ngn2-Isl1-Lhx3 and Bcl-xL in mNSCs lines using a doxycycline (Dox) induction system and concluded that expression of Bcl-xL reduced neuronal death and improved functional maturation of motor neurons. However, even after 7 days of implantation, survival rescue did not appear to be robust and evidence of functional maturation was limited. More importantly, in vivo transplantation did not show significant significance. Other comments are as follows.

1. in summary: "Taken together, these data suggest that genetic modification of stem cells prior to transplantation may improve their survival and increase their function in cell therapy". ". However, there are no supporting in vivo data.

2. "Survival rates were calculated by counting three randomly selected areas per well. The area of the selected area is not known.

3. PI/Hoechst staining is not suitable for apoptosis detection. It is recommended to use FACS and detect Annexin V/PI.

4. The percentage of GFP+ cells should be less than 99.4% according to Fig1 D.

5. In 3.3, why is the starting cell count "similar" and not " the same"?

6. NILB-mNSCs should be compared with NIL-mNSCs in transplantation experiments to show their superiority.

Author Response

Dear reviewer,

We appreciate very much for the careful reading of our manuscript and valuable suggestions. We have carefully considered the comments and have revised the manuscript accordingly.

The following is a point-by-point response to comments, questions and suggestions raised by reviewers.

Comments and Suggestions for Authors

The authors established ectopic expression of Ngn2-Isl1-Lhx3 and Bcl-xL in mNSCs lines using a doxycycline (Dox) induction system and concluded that expression of Bcl-xL reduced neuronal death and improved functional maturation of motor neurons. However, even after 7 days of implantation, survival rescue did not appear to be robust and evidence of functional maturation was limited. More importantly, in vivo transplantation did not show significant significance. Other comments are as follows.

Response: Thanks for the helpful suggestions. We have removed the whole part of in vivo experiments as suggested by Reviewer 1, the issue of cell survival and functions in vivo is now been discussed in the revised manuscript as follow.

“Despite these beneficial potentials of NILB-mNSCs in vitro, the survival and functions of NILB-mNSCs after transplantation in vivo are the most important challenge. Notably, initial loss of graft cells was multifactorial in origin, including immunorejection, anoikis and inflammation-related factors [PMID:PMC2529073, 12518889 and 17721512]. Single interventions had little impact on extending survival of graft cells. The major limitation of our current study is that over express single anti-apoptosis gene Bcl-xL in mNSCs, which has largely failed to extend the survival of these mNSCs after transplantation in vivo. Accordingly, further studies are required to develop a pro-survival cocktail targeting key points of potential death pathways, and determine whether this mNSCs is useful in vivo. Pro-survival strategies have been reported, including genetic modifications, cell preconditioning, and use of biomaterials [PMID: 32518349]. Among them, multigenetic modifications can endow the stem cells with universal capacities, including anti-immune rejection, anti-inflammation, and increasing angiogenesis [PMID: 18031863]. Biomaterials as a supportive matrix for the delivery of cells into the injury cavity protect transplanted stem cells from anoikis [PMCID: PMC4157751]. Therefore, those pro-survival cocktail could be helpful for survival and function of graft.”

  1. in summary: "Taken together, these data suggest that genetic modification of stem cells prior to transplantation may improve their survival and increase their function in cell therapy". ". However, there are no supporting in vivo data.

Response: We are sorry for the inaccurate statement. We have used a more appropriate statement to replace the above statement in the revised manuscript as follow.

“These data indicated that special genetic modification of NSCs will be helpful for survival and formation of motor neurons.”

  1. "Survival rates were calculated by counting three randomly selected areas per well. The area of the selected area is not known.

Response: We have included the cell counting and survival rate calculation in the revised manuscript as follow.

“The surviving cells were quantified at 200× magnification. Three non-overlapping visual fields were randomly selected per well from a 24-well plate. Three independent experiments were performed. The number of surviving neuronal cells was counted, and the average number was calculated for survival rates.”

  1. PI/Hoechst staining is not suitable for apoptosis detection. It is recommended to use FACS and detect Annexin V/PI.

Response: As NILB-mNSCs were GFP positive, Annexin V/PI is not suitable for apoptosis cell detection in our study. So, we referred to previous reports [PMCID: 35943218, PMC7264321] to detect apoptosis cells by PI/Hoechst staining.

  1. The percentage of GFP+ cells should be less than 99.4% according to Fig1 D.

Response: The percentage of GFP+ cells has been re-quantified and corrected in Figure 1D.

  1. In 3.3, why is the starting cell count "similar" and not " the same"?

Response: We are sorry for the inaccurate statement. The same number of NIL-mNSCs and NILB-mNSCs at 5´104/well were planted on Matrigel-coated 24-well plate.

  1. NILB-mNSCs should be compared with NIL-mNSCs in transplantation experiments to show their superiority.

Response: Thanks for the helpful suggestions. As the same with the first question, the issue of cell survival and functions in vivo is now been discussed in the revised manuscript.

“Despite these beneficial potentials of NILB-mNSCs in vitro, the survival and functions of NILB-mNSCs after transplantation in vivo are the most important challenge. Notably, initial loss of graft cells was multifactorial in origin, including immunorejection, anoikis and inflammation-related factors [PMID:PMC2529073, 12518889 and 17721512]. Single interventions had little impact on extending survival of graft cells. The major limitation of our current study is that over express single anti-apoptosis gene Bcl-xL in mNSCs, which has largely failed to extend the survival of these mNSCs after transplantation in vivo. Accordingly, further studies are required to develop a pro-survival cocktail targeting key points of potential death pathways, and determine whether this mNSCs is useful in vivo. Pro-survival strategies have been reported, including genetic modifications, cell preconditioning, and use of biomaterials [PMID: 32518349]. Among them, multigenetic modifications can endow the stem cells with universal capacities, including anti-immune rejection, anti-inflammation, and increasing angiogenesis [PMID: 18031863]. Biomaterials as a supportive matrix for the delivery of cells into the injury cavity protect transplanted stem cells from anoikis [PMCID: PMC4157751]. Therefore, those pro-survival cocktail could be helpful for survival and function of graft.”

Reviewer 3 Report

In this manuscript, the authors first generate Dox-inducible NSC line (NIL-mNSC), differentiating towards motor neurons specifically, then overexpressed Bcl-xL (NILB-mNSC) to further promote its survival. Bcl-xL could not only resist neuronal apoptosis, but also promote the maturation of mNSCs-derived motor neurons. These results are exciting and provide new insights into motor neuron replacement-based treatment of neurodegenerative disorders. In general, the experiments were carefully designed and manuscript is well written, however, some details need to be added.

1. Neuron replacement therapy has been challenged by that transplanted NSC differentiated to glial cells, while not neuron cells. Here, the inducible NIL system overcomes this issue. The authors should introduce more relevant researches about the NSC therapies in the introduction part and cite them properly.

2. In Figure 1C, how long were these cells induced by dox? Around 75% of cells were Isl1/2 positive, the ratio is affected by dox treatment time or not? The authors should include -DOX group as the control.

3. In Figure 3B, 3C and 3E, it is confusing about the labels of NIL-mNSCs and NILB-mNSCs. According to the description in line 206-229, i think the black bar represents NILB-mNSCs, and the white bar represents NIL-mNSCs, right?

4. In Figure S1, the authors showed the survival and differentiation of NILB-mNSCs after transplantation in SOD-G93A mice, did the authors observed some microglia cells in the transplantation site? 

5. Since the microenvironment modulates the differentiation of NSCs, did the authors observed a higher ratio of motor neurons differentiation in figure S1 than in figure 1c, the authors need discuss more about it in the manuscript.

Author Response

Dear reviewer,

We appreciate very much for the careful reading of our manuscript and valuable suggestions. We have carefully considered the comments and have revised the manuscript accordingly.

The following is a point-by-point response to comments, questions and suggestions raised by reviewers.

Comments and Suggestions for Authors

In this manuscript, the authors first generate Dox-inducible NSC line (NIL-mNSC), differentiating towards motor neurons specifically, then overexpressed Bcl-xL (NILB-mNSC) to further promote its survival. Bcl-xL could not only resist neuronal apoptosis, but also promote the maturation of mNSCs-derived motor neurons. These results are exciting and provide new insights into motor neuron replacement-based treatment of neurodegenerative disorders. In general, the experiments were carefully designed and manuscript is well written, however, some details need to be added.

1.Neuron replacement therapy has been challenged by that transplanted NSC differentiated to glial cells, while not neuron cells. Here, the inducible NIL system overcomes this issue. The authors should introduce more relevant researches about the NSC therapies in the introduction part and cite them properly.

Response: Thanks for the suggestions. Sentences were added in the introduction section of revised manuscript to introduce the NSC therapies as follow.

“Several previous studies demonstrated NSCs transplantation improved motor function in different neurodegenerative disease animal models, including ALS [PMID: 16644922], chronic spinal cord injury [PMID: 27485458], Parkinson’s disease and so on [PMID: 23131156]. Nevertheless, there are still many challenges to be faced at the cell biological and neuroscience system levels. First, NSCs are more likely to differentiate into glial cells rather than functional neurons after transplantation in vivo, which is a disadvantage for neuron replacement therapy of neurodegenerative diseases. Tashiro et al. showed that more than 80% NSCs differentiated to glia cells after transplantation in chronic spinal cord injury mice [PMID: 27485458]. Moreover, massive death and functional deficiency of transplanted cells represents a considerable therapeutic hurdle. Ziavra et al. showed that the percentage of surviving NSCs in the host striatum was extremely low (0.2%–0.6%), resulting in limited cell replacement [PMID: 23131156]. The graft cell death was initiated by three principal pathways: immunorejection, involves donor tissue recognition and destruction by allo-specific immune cells of the recipient[PMID: PMC2529073]; anoikis, due to the anchorage-dependent cells detachment from their substrate for injection [PMID: 12518889]; and inflammation-related microenvironment, such as free radicals, cytokines and natural killer cells in the pathological site of neurodegenerative diseases [PMID: 17721512].”

  1. In Figure 1C, how long were these cells induced by dox? Around 75% of cells were Isl1/2 positive, the ratio is affected by dox treatment time or not? The authors should include -DOX group as the control.

Response: In Figure 1C, Ngn2-Isl1-Lhx3 conditional expression was induced with Dox for 48 h in the mNSC medium. Our study and 2 precious studies all showed that a strong induction of the exogenous Ngn2-Isl1-Lhx3 transcript after 24-48 h of Dox addition. Furthermore, long-term expression of exogenous Ngn2-Isl1-Lhx3 is not required for neuronal conversion [PMCID: PMC3843951; 29729503; 35943218]. We have added -Dox group in Figure 1C as suggested in the revised manuscript.

  1. In Figure 3B, 3C and 3E, it is confusing about the labels of NIL-mNSCs and NILB-mNSCs. According to the description in line 206-229, i think the black bar represents NILB-mNSCs, and the white bar represents NIL-mNSCs, right?

Response: We are sorry for the mistake. Here, labels of NIL-mNSCs and NILB-mNSCs have been corrected in Figure 3B, 3C and 3E in the revised manuscript.

  1. In Figure S1, the authors showed the survival and differentiation of NILB-mNSCs after transplantation in SOD-G93A mice, did the authors observed some microglia cells in the transplantation site? 

Response: Yes, a few of glia cells expressed GFAP were also observed in the transplantation sites. However, we have removed the whole part of in vivo experiments as suggested by Reviewer 1, so this data was not shown in the revised manuscript.

  1. Since the microenvironment modulates the differentiation of NSCs, did the authors observed a higher ratio of motor neurons differentiation in figure S1 than in figure 1c, the authors need discuss more about it in the manuscript.

Response: we have removed the whole part of in vivo experiments as suggested by Reviewer 1. In vivo experiments showed NILB-mNSC differentiated into mature MAP+ neurons on day 10 after Dox treatment. This duration was faster than that in previous observations using wild type NPCs, those NSCs differentiated into specific neurons by microenvironment modulation [PMID: 17300870]. Preclinical studies documented that after transplantation in vivo, wild-type NPCs need 1-2 month to form TUJ1+ neuron in ALS animal models [PMID:16644922, 17038899]. Dosage of Dox treatment is also a key point for stem cell differentiation and animal recovery. Overdose of Dox treatment may lead to side effects on gastrointestinal tract [PMID: 3055652]. Otherwise, insufficient induction will make low efficiency of the neuronal differentiation in vivo. Therefore, it is necessary to conduct experiments to determine the optimized dosage and administration route.

Reviewer 4 Report

It may be helpful to the readers if the following are discussed:

1. Did the grafted NSCs survived longer compared to other studies that used NSCs with no BCL expression?

2. What was the possible cause(s) that the grafted cells didn't survive?

Author Response

Dear reviewer,

We appreciate very much for the careful reading of our manuscript and valuable suggestions. We have carefully considered the comments and have revised the manuscript accordingly.

The following is a point-by-point response to comments, questions and suggestions raised by reviewers.

Comments and Suggestions for Authors

It may be helpful to the readers if the following are discussed:

  1. Did the grafted NSCs survived longer compared to other studies that used NSCs with no BCL expression?

Response: Thanks for your comments. Three previous studies also showed that grafted NSCs survived for at least 3 weeks after transplantation in chronic spinal cord injury, Alzheimer’s disease and Parkinson’s disease mouse models, which were longer than that of our Bcl-overexpression NSCs[PMCID: PMC8451553, PMC4971501 and 23131156]. Sentences were added in the discussion section of revised manuscript to discuss graft survival as follow.

“Despite these beneficial potentials of NILB-mNSCs in vitro, the survival and functions of NILB-mNSCs after transplantation in vivo are the most important challenge. Notably, initial loss of graft cells was multifactorial in origin, including immunorejection, anoikis and inflammation-related factors [PMID:PMC2529073, 12518889 and 17721512]. Single interventions had little impact on extending survival of graft cells. The major limitation of our current study is that over express single anti-apoptosis gene Bcl-xL in mNSCs, which has largely failed to extend the survival of these mNSCs after transplantation in vivo. Accordingly, further studies are required to develop a pro-survival cocktail targeting key points of potential death pathways, and determine whether this mNSCs is useful in vivo. Pro-survival strategies have been reported, including genetic modifications, cell preconditioning, and use of biomaterials [PMID: 32518349]. Among them, multigenetic modifications can endow the stem cells with universal capacities, including anti-immune rejection, anti-inflammation, and increasing angiogenesis [PMID: 18031863]. Biomaterials as a supportive matrix for the delivery of cells into the injury cavity protect transplanted stem cells from anoikis [PMCID: PMC4157751]. Therefore, those pro-survival cocktail could be helpful for survival and function of graft.”

  1. What was the possible cause(s) that the grafted cells didn't survive?

Response: The graft cell death was initiated by three principal pathways: immunorejection, involves donor tissue recognition and destruction by allo-specific immune cells of the recipient[PMID:PMC2529073]; anoikis, due to the anchorage-dependent cells detachment from their substrate for injection[PMID:12518889]; and inflammation-related factors, such as free radicals, cytokines and natural killer cells[PMID: 17721512]. These possible causes suggested that multiple, parallel processes were contributing to graft cell death, and that blocking one pathway simply led to cell death by another.

Round 2

Reviewer 2 Report

1. The authors addressed issues partly. 2. Even NILB-mNSCs were GFP positive, Annexin V/PI detection is still available. 3. For the solid evidence, in vivo additional experiments are strongly suggested. 

Author Response

Dear reviewer,

We appreciate very much for the careful reading of our manuscript and valuable suggestions. We have carefully considered the comments and have revised the manuscript accordingly.

The following is a point-by-point response to comments, questions and suggestions raised by reviewers.

  1. The authors addressed issues partly.

Response: Thanks for the comments. We have further improved our manuscript according to requirements.

  1. Even NILB-mNSCs were GFP positive, Annexin V/PI detection is still available.

Response: We have done more experiments to stain differentiated mNSCs with the annexin V and 7-AAD, analyzed by FACS. It showed that about 48.2% cells were 7-AAD and annexin-V dual-positive in NIL-mNSC group. In contrast, lower than 12% cells were positive for both dyes in NILB-mNSC group (Figures 4A–B).

  1. For the solid evidence, in vivo additional experiments are strongly suggested. 

Response: Thanks for the helpful suggestions. Additional in vivo experiments have been done and the results were showed in Figure 5 of revised manuscript. Correspondingly, the results were described as follow.

“3.5. In vivo survival and direct differentiation of NILB-mNSCs induced by Dox.

To trace cells in vivo, NIL-mNSCs and NILB-mNSCs were labelled with EGFP, respectively. mNSCs were intrathecally injected into ALS mice (SOD1-G93A) and Dox was treated every other day (Figure 5A). The grafts survived at least 10 days, the number of GFP positive grafts in NILB-mNSCs group was much more than that of NIL-mNSCs group (Figure 5B). Further immunostaining experiments showed that most grafts expressed neural markers TUJ1, HB9, and MAP2 (Figure 5C).”

Round 3

Reviewer 2 Report

All issues I raised have been addressed by authors.